# Sharing Less is More: Lifelong Learning in Deep Networks with Selective Layer Transfer

## Abstract

Effective lifelong learning across diverse tasks requires diverse knowledge, yet transferring irrelevant knowledge may lead to interference and catastrophic forgetting. In deep networks, transferring the appropriate granularity of knowledge is as important as the transfer mechanism, and must be driven by the relationships among tasks. We first show that the lifelong learning performance of several current deep learning architectures can be significantly improved by transfer at the appropriate layers. We then develop an expectation-maximization (EM) method to automatically select the appropriate transfer configuration and optimize the task network weights. This EM-based selective transfer is highly effective, as demonstrated on three algorithms in several lifelong object classification scenarios.

## 1 Introduction

Transfer at different layers within a deep network corresponds to sharing knowledge between tasks at different levels of abstraction. In multi-task scenarios that involve diverse tasks, reusing low-layer representations may be appropriate for tasks that share feature-based similarities, while sharing high-level representations may be more appropriate for tasks that share more abstract similarities. Selecting the appropriate granularity of knowledge to transfer is an important architectural consideration for deep networks that support multiple tasks.

In scenarios where tasks share substantial similarities, many multi-task methods have found success using a static configuration of the knowledge to share (Caruana, 1997; Yang & Hospedales, 2017; Lee et al., 2019; Liu et al., 2019; Bulat et al., 2020), such as sharing the lower layers of a deep network with upper-level task-specific heads. As tasks become increasingly diverse, the appropriate granularity for transfer may vary between tasks based on their relationships, necessitating more selective transfer. Prior work in selective sharing for deep networks has typically either (1) branched the network into a tree structure (Lu et al., 2017; Yoon et al., 2018; Vandenhende et al., 2019; He et al., 2018), which emphasizes the sharing of lower layers or (2) introduced new learning modules between task models (Yang & Hospedales, 2017; Xiao et al., 2018; Cao et al., 2018) which increases the complexity of training. The transfer configuration could then be optimized in batch settings to maximize performance across the tasks.

However, the problem of selective transfer is further compounded in continual or lifelong learning settings, in which tasks are presented consecutively. The optimal transfer configuration may vary between tasks or it may vary over time. And indeed, we may *not* want to transfer at all layers, as some task-specific layers may need to be interleaved with shared knowledge in order to customize that shared knowledge to individual tasks. To verify this premise and motivate our work, we conducted a simple brute-force initial experiment: we took a multi-task CNN with shared layers and a lifelong learning CNN that uses factorized transfer (DF-CNN, Lee et al., 2019) and varied the set of CNN layers that employed transfer (with task-specific fully connected layers at the top). Using two data sets, we considered several transfer static configurations: transferring at all CNN layers, transfer at the top-$k$ CNN layers, transfer at the bottom-$k$ CNN layers, and alternating transfer/no-transfer CNN layers. The results are shown in Figure 1, with details given in Section 2. Clearly, we see that the optimal *a posteriori* transfer configuration varies between task relationships and transfer mechanisms. Restricting the transfer layers significantly improves performance over the naïve approach of transferring at all layers, with the alternating configuration performing extremely well for both multi-task and lifelong learning.

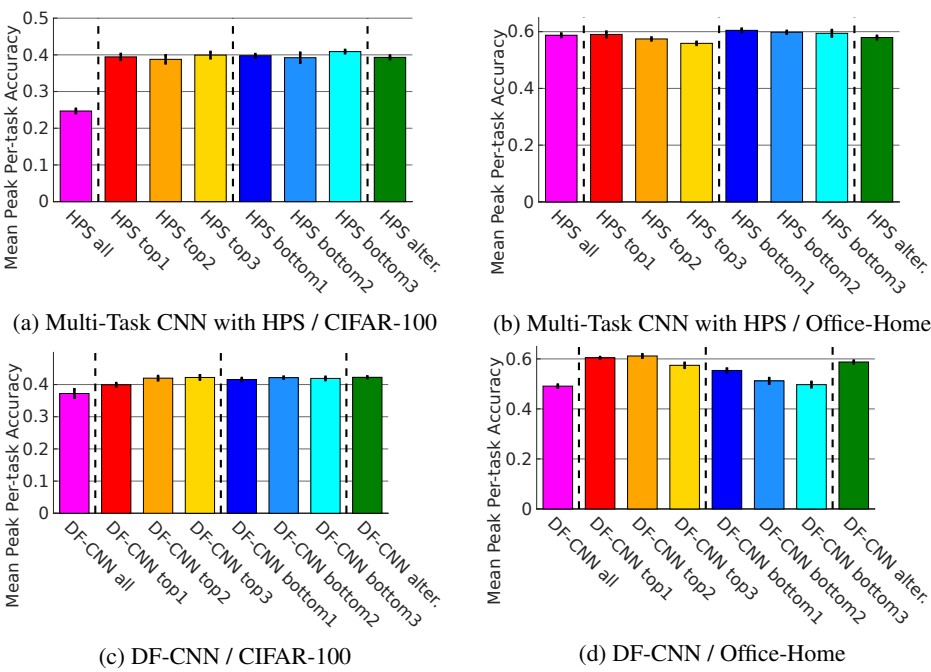

Figure 1: Accuracy of CNN models averaged over ten tasks in a lifelong learning setting with 95% confidence interval. This empirically shows that the optimal transfer configuration varies, and choosing the correct configuration is superior to transfer at all layers.

Instead of only considering such a restricted set of static configurations in brute-force search, our goal is automate this process of selective transfer during learning, enabling it to customize the transfer configuration to each task. We investigate the use of architecture search during learning to dynamically adjust the transfer configuration between tasks and over time, using expectation-maximization (EM) to learn both the parameters of the task models and the layers to transfer within the deep net. This approach, Lifelong Architecture Search via EM (LASEM), enables deep nets to transfer different sets of layers for each task, allowing more flexibility over prior work in branching-based configurations for selective transfer. It also introduces little additional computational cost over the base learner in comparison to training selective transfer modules between task networks, and in contrast to the expense of brute-force search over all transfer configurations. To demonstrate its effectiveness, we applied LASEM to three architectures that support transfer between tasks in several lifelong learning scenarios and compared it against other lifelong learning and architecture search methods.

## 2 THE EFFECT OF DIFFERENT TRANSFER CONFIGURATIONS

This section further describes the initial experiments mentioned in the introduction as motivation for our proposed LASEM method. The hypothesis of our work is that lifelong deep learning can benefit from using a more flexible transfer mechanism that selectively chooses the transfer architecture configuration for each task. This would permit it to dynamically select, for each task model, which layers to transfer and which to keep as task-specific (enabling it to customize transferred knowledge to an individual task).

To determine the effect of different transfer configurations, we conducted a set of initial experiments using two established methods:

**Multi-Task CNN with hard parameter sharing (HPS):** This approach shares the hidden CNN layers between all tasks, and maintains task-specific fully connected output layers. It is one of the most common methods for multi-task learning of neural nets (Caruana, 1997), and is widely used.

**Deconvolutional factorized CNN (DF-CNN):** The DF-CNN (Lee et al., 2019) adapts CNNs to a continual learning setting by sharing layer-wise knowledge across tasks. Instead of using the same

convolutional filters for multiple tasks, the convolutional filters are dynamically generated from a task-*independent* layer-*dependent* shared tensor through a task-specific deconvolutional operation and tensor contraction. Similar to HPS, the DF-CNN maintains task-specific fully connected topmost layers. When training the task models consecutively, gradients flow through to update the shared tensors and the task-specific parameters that transform those shared tensors to construct the task CNN. This transfer architecture enables the DF-CNN to learn and compress knowledge universal among the observed tasks into the shared tensors.

Both these methods utilize a set of transfer-based CNN layers and non-transfer task-specific layers. For a network with $d$ CNN layers, there are $2^d$ potential transfer configurations. To explore the effect of transfer at different layers, we varied the transfer configuration among several options:

- *All*: Transfer at all CNN layers. Note that the original DF-CNN used this configuration.
- *Top $k$*: Transfer across task models occurs only at the $k$ highest CNN layers, with all others remaining task-specific. We would expect this transfer configuration to benefit tasks that share high-level concepts but have low-level feature differences.
- *Bottom $k$*: Transfer occurs only at the $k$ lowest CNN layers, with all others remaining task-specific. This architecture is opposite of the *Top $d - k$*, so we would expect it to benefit tasks that share perceptual similarities but have high-level differences.
- *Alternating*: This configuration alternates transfer and non-transfer layers, enabling the non-transfer task-specific layers to further customize the transferred knowledge to the task.

We evaluated the performance of various transfer configurations on the CIFAR-100 (Krizhevsky & Hinton, 2009) and Office-Home (Venkateswara et al., 2017) data sets, following the lifelong learning experimental setup used in previous work (Lee et al., 2019). CIFAR-100 involves ten consecutive tasks of ten-way image classification, where any object class occurs in only one task. Office-Home involves ten tasks of thirteen-way classification, separated into two domains: "Product" images and "Real World" images. The CNN architectures used for each data set and optimization settings follow prior work and are detailed in Appendix A. During training, we measured the peak per-task accuracy on held-out test data, averaging results over five trials.

Our results in Figure 1 reveal that permitting transfer at all layers does not guarantee the best performance. This observation complicates learning on novel tasks, since the best transfer configuration depends both on the algorithm and the task relations in the data set. Notably, we see that the DF-CNN, which is designed for lifelong learning, can be improved beyond the original version (Lee et al., 2019) by allowing transfer at only some layers. Furthermore, we can see that the optimal transfer configuration varies between data sets and algorithms. For instance on Office-Home, sharing lower layers in the HPS multi-task CNN achieves better performance on average, but transferring upper layers works better for the DF-CNN. Similarly, the Alternating configuration consistently achieves near the best performance for the DF-CNN, benefiting from permitting the non-transfer layers to customize transferred knowledge to the individual task, but it is not consistently as good for HPS.

## 3 ARCHITECTURE SEARCH FOR THE OPTIMAL TRANSFER CONFIGURATION

The experiment presented above reveals that the transfer configuration can have a significant effect on lifelong learning performance, and that the best transfer configuration varies. These observations inspire our work to develop a more flexible mechanism for selective transfer in lifelong learning.

We can view the transfer configuration as a new hyper-parameter for each task model. Even with the constraint that all task models use the same transfer configuration, the search space grows exponentially as the neural network gets deeper (i.e., $2^d$ configurations for $d$ CNN layers). In more flexible settings where the transfer configuration is customized to each task, this search space grows even more, linearly in the number of tasks. Although we could compound this problem further by permitting partial transfer *within* a layer, we focus on optimizing layer-based transfer configurations.

Formally, a layer-based transfer configuration for task $t$ can be specified by a $d$-dimensional binary vector $c_t \in \mathcal{C} = \{0, 1\}^d$, where each $c_{t,j}$ is a binary indicator whether or not the $j$th layer involves transfer. We can compactly notate $c_t$ by a set of indices of transferred layers. For example, the Alternating configuration described in Section 2 can be denoted by $c = [0, 1, 0, 1] = \{2, 4\}$; Figure 2 depicts this particular configuration for three approaches.

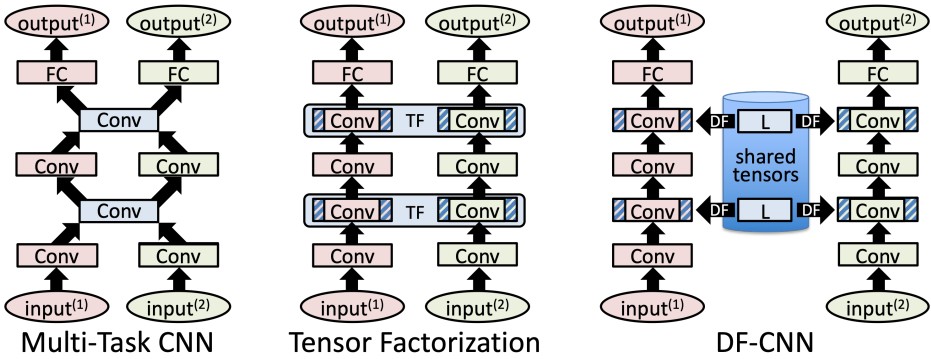

Figure 2: The Alternating $\{2, 4\}$ transfer configuration for three different approaches using CNNs with four convolutional layers and one fully connected layer. Models are illustrated for two tasks, red and green, with shared or transfer-based layers denoted in blue.

Our goal is to determine the task-specific transfer configuration while simultaneously optimizing the log-likelihood of the task models and shared knowledge in a lifelong setting. Treating $c_t$ as a latent variable of the model for task $t$, we can employ expectation-maximization (EM) to perform this joint optimization. For each layer $l$, LASEM maintains a shared set of transfer-based model parameters $\boldsymbol{\theta}_s^{(l)}$ and, for each new task, a set of task-specific model parameters $\boldsymbol{\theta}_t^{(l)}$, using the chosen configuration $\boldsymbol{c}_{(t)}$ to determine which combination of parameters will be used to form the specific model for that task. In brief, the E-step estimates the usefulness of the representation that each transfer configuration $\boldsymbol{c}_i \in \mathcal{C}$ can learn from the given data (i.e., the likelihood of data $P(y_{new} \mid X_{new}, \boldsymbol{c}_i)$), while the M-step optimizes parameters of the task model and shared knowledge. We next detail this approach.

We first consider how to model the prior $\pi_t$ on possible configurations of the current task's $\boldsymbol{c}_{(t)}$. Using a simple frequency-based probability estimate with Laplace smoothing represents the prior probability of each transfer configuration as

$$P(\boldsymbol{c}_{(t)} = \boldsymbol{c}_i) = \pi_t(\boldsymbol{c}_i) = \frac{n_{\boldsymbol{c}_i} + 1}{\sum_j (n_{\boldsymbol{c}_j} + 1)} \quad , \tag{1}$$

where $\boldsymbol{c}_{(t)}$ denotes the configuration for task $t$, and $n_{\boldsymbol{c}_i}$ is the number of previous mini-batches for which $\boldsymbol{c}_i$ is the most probable configuration. This estimate, which we adopt in the experiments, considers each transfer configuration solely based on the current task's data. Alternative priors could instead be used, such as measuring the historic transfer configuration frequency over all tasks (which assumes substantial similarity among tasks) or measuring configuration frequency over related tasks (which requires a notion of task similarity, such as via a task descriptor (Isele et al., 2016; Sinapov et al., 2015), or as determined dynamically by the task model's relation to shared knowledge). During development, we also considered estimating the prior based on the probability of configurations averaged over training samples instead of the number of mini-batches $n_{\boldsymbol{c}_j}$, but this alternative was not different statistically from Equation 1 in an empirical evaluation on CIFAR100 and Office-Home.

In the E-step, the posterior on configurations is derived by combining the above prior and likelihood, which can be computed from the output of the task network on the current task's data $(X_{new}, y_{new})$:

$$P(\boldsymbol{c}_i \mid X_{new}, y_{new}) \propto P(\boldsymbol{c}_{(t)} = \boldsymbol{c}_i) P(y_{new} \mid X_{new}, \boldsymbol{c}_i) \quad . \tag{2}$$

The M-step improves the log-likelihood via the estimated probability distribution over the transfer configurations. Both $\boldsymbol{\theta}_s$ and $\boldsymbol{\theta}_t$ are updated by the aggregated gradients of the log-likelihood in cases where the transfer configurations match the corresponding parameter. To combine the gradients of a specific parameter vector ($\boldsymbol{\theta}_s$ and $\boldsymbol{\theta}_t$) over multiple possible configurations, we take the sum of the corresponding gradients weighted by the posterior estimate in Equation 2.

$$\begin{aligned}
\boldsymbol{\theta}_s^{(l)} &\leftarrow \boldsymbol{\theta}_s^{(l)} + \lambda \sum_{i:c_{i,l}=1} P(\boldsymbol{c}_i \mid X_{new}, y_{new}) \nabla \log P(y_{new} \mid X_{new}, \boldsymbol{c}_i) \quad \forall l \in \{1, \cdots, d\} \\
\boldsymbol{\theta}_t^{(l)} &\leftarrow \boldsymbol{\theta}_t^{(l)} + \lambda \sum_{i:c_{i,l}=0} P(\boldsymbol{c}_i \mid X_{new}, y_{new}) \nabla \log P(y_{new} \mid X_{new}, \boldsymbol{c}_i) \quad \forall l \in \{1, \cdots, d\}
\end{aligned} \tag{3}$$

---

**Algorithm 1** Lifelong Learning with EM-based Partial Layer Transfer

1: $\{\boldsymbol{\theta}_s^{(l)}\}_{l=1}^d \leftarrow$ randomInitializer(netSize)
2: **while** isMoreTrainingDataAvailable() **do**            ▷ Loop over entire training lifetime
3:      $(X_{new}, y_{new}, t) \leftarrow$ getNextTrainingData()            ▷ Get current task data
4:      **if** isNewTask($t$) **then**            ▷ Initialize parameters for new tasks
5:          $\{\boldsymbol{\theta}_t^{(l)}\}_{l=1}^d \leftarrow$ randomInitializer(netSize)
6:          $\pi_t \leftarrow$ priorInitializer()
7:      **end if**
8:      Calculate likelihood $P(y_{new} \mid X_{new}, \boldsymbol{c}_i) \,\forall \boldsymbol{c}_i \in \mathcal{C}$
9:      Calculate posterior $P(\boldsymbol{c}_i \mid X_{new}, y_{new}) \propto \pi_t(\boldsymbol{c}_i) P(y_{new} \mid X_{new}, \boldsymbol{c}_i) \,\forall \boldsymbol{c}_i \in \mathcal{C}$    ▷ Eqn 2
10:      **loop** $numMSteps$ **times**            ▷ Eqn 3
11:          $\{\boldsymbol{\theta}_s^{(l)}, \boldsymbol{\theta}_t^{(l)}\}_{l=1}^d \leftarrow$ gradOptimizer$\Big(X_{new}, y_{new}, \lambda, P(\mathcal{C} \mid X_{new}, y_{new}), \{\boldsymbol{\theta}_s^{(l)}, \boldsymbol{\theta}_t^{(l)}\}_{l=1}^d\Big)$
12:      **end loop**
13:      $\pi_t \leftarrow$ priorUpdater($\pi_t, P(c_i \mid X_{new}, y_{new})$)            ▷ Eqn 1
14: **end while**

---

The main difference in the update rules in Equation 3 is the condition for the index of the summation. Since one gradient step on the configurations $\{\boldsymbol{\theta}_s^{(l)}, \boldsymbol{\theta}_t^{(l)}\}_{l=1}^d$ may have little effect on the likelihood, we can hold the likelihood fixed to take multiple M-steps per E-step by iterating Equation 3.

Our approach, Lifelong Architecture Search via EM (LASEM), is specified in Algorithm 1. The parameters of the transfer-based components are initialized only at the beginning of the lifelong learning process (line 1), while the parameters of the task-specific components and prior probability of configurations are initialized when the algorithm encounters a new task (lines 4-7). Each iteration, the lifelong learner obtains a mini-batch of labeled data $(X_{new}, y_{new})$ drawn i.i.d for some task $t$, training the task model online. Typically, there would be multiple consecutive mini-batches experienced per task before the environment would switch to a new task. The algorithm applies the E-step (lines 8, 9, and 13) and M-step (lines 10–12) using the data mini-batch at each iteration. LASEM takes E- and M-steps for each mini-batch, so over the consecutive mini-batches per task, this process would be similar to the alternation of multiple E- and M-steps performed by the standard EM algorithm. We consider lifelong learning scenarios in which a learner has no control over tasks, so the current task can be switched to another one without convergence of the learning algorithm. Because of this setting, we do not require EM convergence, but it is possible to monitor the convergence by checking the probability weight over transfer configurations.

LASEM uses one set of transfer-based and task-specific parameters ($\boldsymbol{\theta}_s$ and $\boldsymbol{\theta}_t$) for *all* transfer configurations, rather than maintaining distinct sets of parameters for each configuration. This approximation reduces the number of parameters and permits parameter updates across transfer configurations via a single gradient step.

*Computational Cost*     The computational cost of LASEM depends heavily on the tasks, the choice of deep net, and how quickly the transfer configuration converges. We show empirically in Section 4.2 that LASEM in practice introduces relatively little additional time complexity ($\sim$30–50% over the base learner's time). Empirically, the E-step takes $\sim$15–20% time of the M-step, but frequent switching of the configuration by the E-step may interfere with the M-step, consequently harming the convergence speed. Taking more M-steps per E-step (by increasing $numMSteps$) can improve this stability and consequently the computational complexity by accelerating convergence.

For a $d$-layer neural network with a time complexity of $N(\cdot)$, the per-iteration computational complexity of both the E- and M-steps are $O(2^d N(\cdot))$. As the network depth $d$ increases, it is well known that neural architecture search (NAS) (Pham et al., 2018; Liu et al., 2018) requires exponentially additional computation. To remedy this issue, LASEM can adopt a similar solution to that of these other NAS methods by considering transfer configurations over *groups* of layers instead of individual layers; we explore this variation in Appendix E. Memory requirements are analyzed in Appendix F.

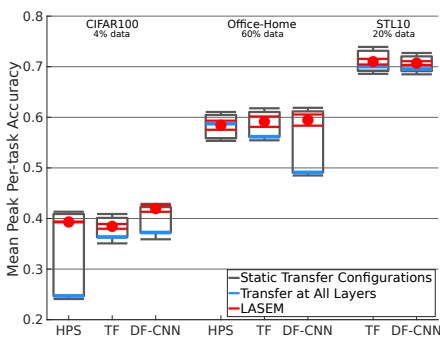 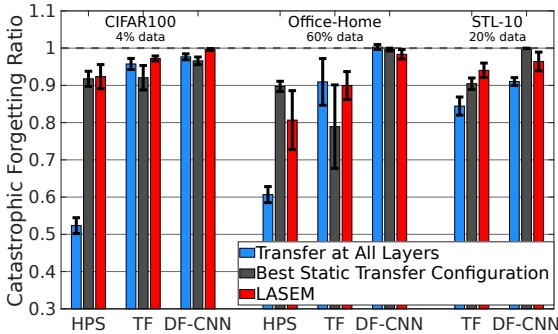

Figure 3: (Left) Performance of LASEM applied to three methods and three lifelong scenarios. Black boxes show the range of mean accuracies that different static configurations can achieve, with the blue lines denoting mean performance of Transfer at All Layers. The red dots denote the mean performance of LASEM. Whiskers depict 95% confidence intervals. (Right) Mean catastrophic forgetting ratio after training all tasks. Less forgetting is indicated by a ratio near 1.0; see Appendix B.

# 4 EXPERIMENTS

We evaluated LASEM following the same experimental protocol for lifelong learning as used in Section 2; see Appendix A for details. In addition to using the CIFAR100 and Office-Home data sets, we introduce a lifelong learning version of the STL-10 data set (Coates et al., 2011). STL-10 has 5,000 training and 8,000 test images divided evenly among 10 classes, with higher resolution than CIFAR-100. We constructed 20 tasks of three-way classification using 20% and 5% of the given training data for training and validation, respectively. To increase the task variations, for each task we randomly chose three image classes, applied Gaussian noise to the images with a random mean and variance, and randomly permuted the channels. All results were averaged over five trials with different random seeds. The code and data set generators are available at http://bit.ly/ICLR_LASEM.

## 4.1 PERFORMANCE OF LASEM

We applied LASEM to three algorithms: a multi-task CNN with hard-parameter sharing (HPS) (Caruana, 1997), Tensor Factorization (TF) (Yang & Hospedales, 2017; Bulat et al., 2020) and the Deconvolutional Factorized CNN (DF-CNN) (Lee et al., 2019). HPS interconnects CNNs in tree structures, with task models explicitly using the same parameters of all shared layers. In contrast, the TF and DF-CNN task models explicitly share only a fraction of tensors, and the parameters of each task model are generated via transfer.

Figure 3 (left) compares the performance of the task-specific transfer configurations discovered by LASEM (red) to using a single static transfer configuration (black boxes). These black boxes depict the performance range of the methods using various static transfer configurations (i.e., All, Top $k$, Bottom $k$, Alternating) for all task models, with All shown in blue. To estimate this range, we tested eight (50%) and 16 (25%) of the possible static configurations for the four-CNN-layer (CIFAR-100 and Office-Home) and six-CNN-layer (STL-10) task models, respectively.

We can see that LASEM chose transfer configurations that perform toward the top of each range, especially on the DF-CNN designed for lifelong learning. LASEM clearly outperforms Transfer at All Layers. Automatically selecting the transfer configuration becomes even more beneficial for methods that have a wide range of performances for different configurations. Examining the catastrophic forgetting ratio (Figure 3 right, with details in Appendix B) reveals the importance of selecting the appropriate transfer configuration for maintaining performance on previous tasks, revealing that LASEM exhibited less forgetting than baselines in most cases, especially on the DF-CNN. Moreover, LASEM imposes little additional cost in order to determine the transfer configuration. In timing experiments, we found that, compared to training with a pre-determined static configuration, LASEM requires only 30-50% additional wall-clock time to search over 16 configurations of a network with four layers, and only double the time to search over 64 configurations of a network with six layers. In

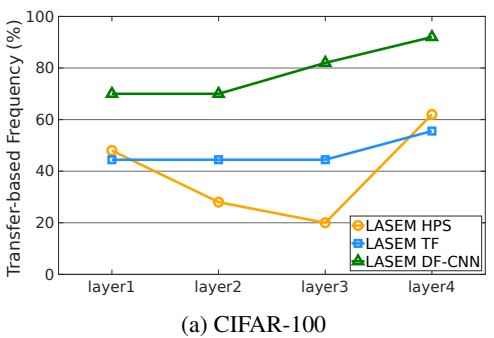
(a) CIFAR-100

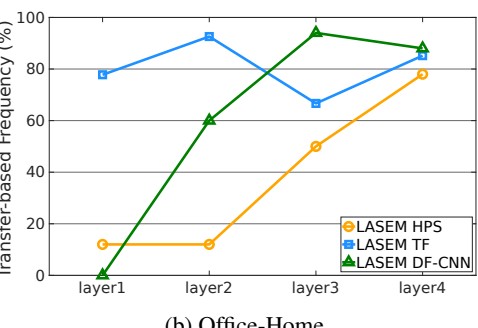
(b) Office-Home

Figure 4: Frequency of each layer being transfer-based according to the selection of LASEM. Generally, upper layers are preferable for transfer, but there are exceptions, i.e. HPS on CIFAR-100.

| Selective Sharing | Accuracy(%) | Forgetting Ratio | Time (k sec) |
|---|---|---|---|
| DEN | $48.00 \pm 0.60$ | $0.28 \pm 0.01$ | 55.9 |
| APD-Net | $\textbf{59.58} \pm \textbf{0.45}$ | $0.83 \pm 0.03$ | 21.5 |
| ProgNN | $\textbf{60.03} \pm \textbf{0.45}$ | $1.00 \pm 0.00$ | 96.7 |
| DARTS HPS | $45.64 \pm 1.20$ | $0.70 \pm 0.07$ | 43.8 |
| DARTS DF-CNN | $56.77 \pm 0.49$ | $0.35 \pm 0.04$ | 33.2 |
| LASEM HPS | $58.44 \pm 0.90$ | $0.81 \pm 0.08$ | 70.2 |
| LASEM TF | $59.14 \pm 0.80$ | $0.90 \pm 0.04$ | 77.3 |
| LASEM DF-CNN | $\textbf{59.45} \pm \textbf{1.10}$ | $0.98 \pm 0.01$ | 83.2 |

Table 1: Comparison of peak per-task accuracy, forgetting, and training time for the same epochs between baselines and LASEM on Office-Home, $\pm$ 95% confidence interval.

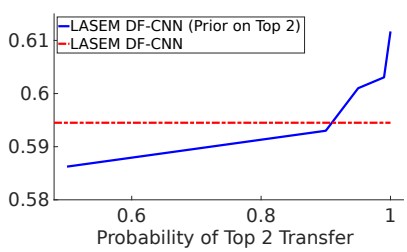

Figure 5: Performance of LASEM DF-CNN compared to LASEM with a fixed posterior distribution over the optimal configuration (on Office-Home).

stark contrast, brute-force search over the transfer configurations requires $15\times$ and $63\times$ *additional* time per task, respectively, over the base learner (see Appendix C).

The ratio of transfer configurations chosen by LASEM are depicted in Figure 4; see Appendix D for detailed results. Figure 4 shows the proportion of time each layer was chosen to be transfer-based. We see that HPS tends to prefer task-specific layers, while TF and DF-CNN are more likely to use transfer layers due to their ability to adapt transferred knowledge. We can also see trends among the chosen layers, such as DF-CNN preferring transfer among higher layers.

## 4.2 COMPARISON TO OTHER SELECTIVE TRANSFER ALGORITHMS

We compared the performance of LASEM on Office-Home against other methods that employ some notion of selective transfer, including the Dynamically Expandable Network (DEN) (Yoon et al., 2018), the Additive Parameter Decomposition Network (APD-Net) (Yoon et al., 2020), the Progressive Neural Net (ProgNN) (Rusu et al., 2016), and Differentiable Architecture Search (DARTS) (Liu et al., 2018). DEN is a lifelong learning architecture that extends HPS by expanding, splitting, and selectively retraining the network to introduce both shared and task-specific parameters in each layer if required. APD-Net has base parameters shared across tasks like HPS, but introduces task-specific masks and additive parameters for adaptation to each task. ProgNN learns lateral connections from earlier task models to the current task model. Both DEN and ProgNN can support complex transfer configurations due to their lack of constraints, such as no assumption of a tree-structured configuration. For example, a ProgNN with all zero-weighted lateral connections for a level creates a task-specific layer, and zero-weighted current task model connections creates a transfer-based layer. DARTS is another general framework for neural architecture search, determining both the most suitable operation of each layer and the best architecture of stacking these layers simultaneously.

Table 1 summarizes the performance of these methods and our approach: mean peak per-task accuracy, catastrophic forgetting ratio and training time. APD-Net, ProgNN and LASEM DF-CNN

are statistically indistinguishable and perform better than the other methods in terms of accuracy. However, APD-Net is weaker in retaining the knowledge of earlier tasks, as shown by its catastrophic forgetting ratio being significantly lower than ProgNN and LASEM DF-CNN. LASEM DF-CNN is ~14% faster than ProgNN, whose time complexity is proportional to the square of the number of tasks. DEN and DARTS have better training times, but fail to perform as well. Note that LASEM shows high accuracy regardless of the base lifelong learner (e.g., HPS, TF, or DF-CNN) while introducing relatively little additional time complexity (~30–50% over the base learner's time).

### 4.3 THE EFFECT OF NON-OPTIMAL TRANSFER CONFIGURATIONS ON LASEM

Besides the capability to customize the transfer configuration to each task, LASEM has a key difference from using a static transfer configuration as in Section 2: LASEM updates both the transfer-based and task-specific parameters ($\boldsymbol{\theta}_s$ and $\boldsymbol{\theta}_t$) by gradients backpropagated from the loss of *all* configurations weighted by the posterior. Consequently, gradients from non-optimal configurations might act as noise or be counterproductive to the optimization process.

To determine whether this aspect had a significant effect on LASEM, we performed an ablative experiment using a static probability on the transfer configurations, instead of the posterior derived from data. This makes LASEM always select the same transfer configuration for all task models, with the experiment varying the noise and controlling the amount of adverse effects from non-optimal configurations during LASEM's optimization.

We first determined the optimal transfer configuration (Top 2 in this experiment), and gave it a static probability of selection, which we varied from $P = 0.5$ to $1$ with a uniform distribution over other configurations. Figure 5 compares the full LASEM DF-CNN against the ablated version. Knowing the correct transfer configuration *a priori* (when $P = 1$) certainly does improve performance, as we would expect, but the overall performance difference as $P$ varies is relatively small. Therefore, the effect of interference from considering non-optimal configurations is minimal, but does exist. Using a more informed prior over the transfer configurations, such as initializing it from the posteriors from previously learned related tasks, may further improve LASEM, which we leave to future work.

## 5 RELATED WORK

The simplest transfer mechanism is hard parameter sharing (HPS), which directly reuses parameters (e.g., layers) between task models (Caruana, 1997). HPS is beneficial when tasks share identical features, but its structural rigidity degenerates as tasks become diverse. Constraints such as regularization (Kirkpatrick et al., 2017; Yoon et al., 2018; He et al., 2018), orthogonality (Suteu & Guo, 2019; Riemer et al., 2019; Farajtabar et al., 2019) or attention mechanisms (Serra et al., 2018; Yoon et al., 2020; Abati et al., 2020) may reduce interference among tasks, but can deter positive transfer. Using tree-like structures (Lu et al., 2017; Vandenhende et al., 2019; He et al., 2018) as the transfer configuration for HPS in multi-task nets give flexibility, but assume that lower level representations are shared, which may not be the case for diverse tasks, as shown in this paper.

Soft parameter sharing (Duong et al., 2015; Bilen & Vedaldi, 2017) builds task-specific networks with weights that are related to other task models via implicit constraints. This architecture provides flexibility to the representations that each task network can learn, so it typically outperforms HPS for more diverse tasks. Success has often been found by using task-agnostic shared knowledge with a task-specific mapping from that shared knowledge to the task models, facilitating transfer between tasks (Yang & Hospedales, 2017; Bulat et al., 2020; Lee et al., 2019; Liu et al., 2019). These works focus on the mapping operation, but put less importance on what layers to transfer, as we explored.

Direct reuse of learned representations from previous tasks models (Rusu et al., 2016; Misra et al., 2016; Cao et al., 2018; Xiao et al., 2018) prevents forgetting, but only permits forward transfer to new tasks (not reverse transfer) and exhibits super-linear training time w.r.t the number of tasks. Progress and compress (Schwarz et al., 2018) tackles this issue by combining progressive neural nets and elastic weight consolidation (Kirkpatrick et al., 2017), but this method has a similar capacity issue of HPS for diverse tasks since one neural net must handle all learned tasks.

Neural architecture search (NAS) examines both the operators and their order in a neural net to optimize performance (Elsken et al., 2018). Our problem of selective layer-based transfer is an

instance of NAS. Strategies for NAS include reinforcement learning (Tan et al., 2019; Chang et al., 2019), evolutionary algorithms (Fernando et al., 2017) and gradient-based learning (Alet et al., 2018). In contrast to these methods, DARTS makes optimization more feasible by using a soft selection of the operators, a weighted sum of operations. Most NAS methods including DARTS train better once the architecture has stabilized, but the weights of DARTS' soft selections are susceptible to vanishing gradients, so it is slower to stabilize than LASEM.

## 6 CONCLUSION

We have shown that the transfer configuration can have a significant impact on lifelong learning, and that the configuration can be dynamically selected during the lifelong learning process with minimal computational cost. Choosing the optimal transfer configuration significantly improves the performance of the DF-CNN and TF over the original method. Using a dynamic transfer configuration reduces the assumptions of algorithm designers in terms of task similarities, but opens potential for the selected configurations to incorporate biases from data.

Although we focused on layer-based transfer, LASEM could easily be extended to support partial transfer within a layer by imposing within-layer partitions and redefining the transfer configuration space $C$ to support those partitions. Discovering these partitions directly from data, or providing more flexible mechanisms for partial within-layer transfer may further improve performance.

ACKNOWLEDGMENTS

Omitted for blind review.

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

Supplementary Material for ICLR 2021 Submission
# Sharing Less is More: Lifelong Learning in Deep Networks with Selective Layer Transfer

## A  EXPERIMENT DETAILS

This section provides detail on the experiments from the main paper. The experiments are based on three image recognition datasets: CIFAR-100 (Krizhevsky & Hinton, 2009), Office-Home (Venkateswara et al., 2017) and STL-10 (Coates et al., 2011).

CIFAR-100 consists of images of 100 classes. The lifelong learning tasks are created following Lee et al. (2019) by separating the dataset into ten disjoint sets of ten classes, and randomly selecting 4% of the original training data to generate training and validation sets in the ratio of 5.6:1 (170 training and 30 validation instances per task). The images are used after normalization.

The Office-Home dataset has images of 65 classes in four domains. Again following Lee et al., lifelong learning tasks are generated by choosing ten disjoint groups of thirteen classes in two domains: Product and Real-World. There is no pre-defined training/testing split in Office-Home, so we randomly split the images in the ratio 6:1:3 for the training, validation, and test sets. The image sizes are not uniform, so we resized all images to be 128-by-128 pixels and re-scaled each pixel value to the range of $[0, 1]$.

We introduce a lifelong learning variant of the STL-10 dataset, which contains ten classes. We constructed 20 three-way classification tasks by randomly choosing the classes, applying Gaussian noise to the images (with a mean and variance randomly sampled from $\{-10\%, -5\%, 0\%, 5\%, 10\%\}$ of the range of pixel values) after re-scaling each pixel value to the range of $[-0.5, 0.5]$, and randomly swapping channels. Note that any pair of tasks differs by at least one image class, the mean and variance of the Gaussian noise, or the order of channels for the swap. We sampled 25% of the given training data and split it into training and validation sets with the ratio 5.7:1 (318 training and 57 validation instances per task). All of the original STL-10 test data are used for held-out evaluation of performance.

The architectural details of the task models used for each data set are described in Figure 6. We used the following values for the hyper-parameters of the algorithms, following the original papers wherever possible:

- The multi-task CNN with hard parameter sharing (HPS) has no additional hyper-parameters.

- Tensor factorization has a scale of the weight orthogonality constraint, whose value was chosen by grid search among $\{0.001, 0.005, 0.01, 0.05, 0.1\}$ following the original paper (Bulat et al., 2020).

| Dataset | CIFAR-100 | Office-Home | STL-10 |
|---|---|---|---|
| Number of Tasks | 10 | 10 | 20 |
| Type of Task | Heterogeneous Classification | | Semi-heterogeneous Classification |
| Classes per Task | 10 | 10 | 3 |
| Amount of Training Data | 4% | - | 25% |
| Ratio of Training and Validation Set | 5.6:1 | 6:1 | 5.7:1 |
| Size of Image | $32 \times 32$ | $128 \times 128$ | $96 \times 96$ |
| Optimizer | RMS Prop | | |
| Learning Rate | $1 \times 10^{-4}$ | $2 \times 10^{-5}$ | $1 \times 10^{-4}$ |
| Epoch per Task | 2000 | 1000 | 500 |
| Ratio of M-steps to E-step ($numMSteps$) | 1 | | |

Table 2: Parameters of the lifelong learning experiments. We used the notion of task type (e.g. heterogeneous) introduced in previous work of multi-task learning (Yang & Hospedales, 2017), which is based on the similarity of data distribution of tasks.

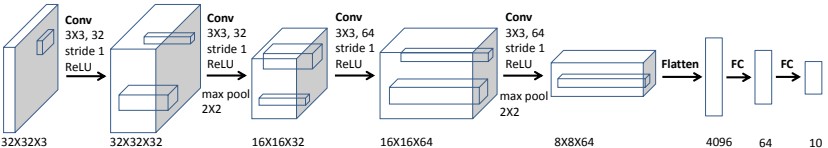

(a) Architecture of task models of CIFAR-100 experiment

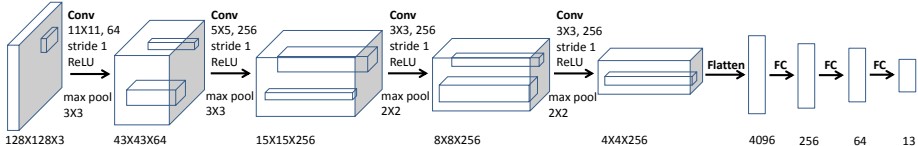

(b) Architecture of task models of Office-Home experiment

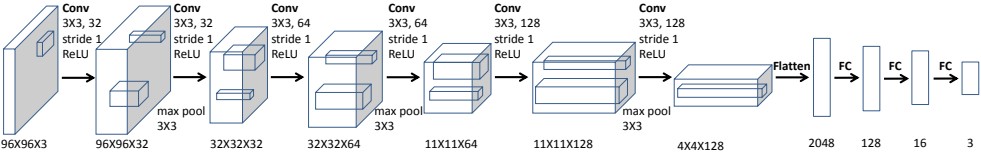

(c) Architecture of task models of STL-10 experiment

Figure 6: Details of the task model architectures used in the experiments. Text by each convolutional layer describes the filter sizes and the number of channels. All convolutional layers are zero-padded.

- DF-CNN requires the size of the shared tensors and the parameters of the task-specific mappings to be specified. Following the original paper (Lee et al., 2019), we chose the spatial size of the shared tensors to be half the spatial size of the convolutional filters, and the spatial size of the deconvolutional filters as $3 \times 3$. For each convolutional layer with input channels $c_{in}$ and output channels $c_{out}$, the number of channels in the shared tensors was one-third of $c_{in} + c_{out}$ and the number of output channels of the deconvolutional filters was two-thirds of $c_{in} + c_{out}$.

- DEN has several regularization terms and the size of the dynamic expansion. We used the regularization values in the authors' published code, and set the size of the dynamic expansion to be 32 by choosing the most favorable value among $\{8, 16, 32, 64\}$.

- APD-Net has two regularization terms for the sparsity of additive parameters $\lambda_1$ and catastrophic forgetting $\lambda_2$. As described in the original paper (Yoon et al., 2020), we used $4e^{-4}$ and 100 as the value of $\lambda_1$ and $\lambda_2$, respectively.

- ProgNN requires the compression ratio of the lateral connections, which we set to be 2, following the original paper (Rusu et al., 2016).

- For DARTS, we used the hyper-parameter settings described in the original paper (Liu et al., 2018).

A lifelong learner has access to the training data of only the current task, and it optimizes the parameters of the current task model as well as any shared knowledge, depending on the algorithm. After the pre-determined number of training epochs, the task switches to a new one regardless of the convergence of the lifelong learner, which favors learners that can rapidly adapt to each task. When the learner encounters a new task, it initializes newly introduced parameters of the new task model, but re-uses the parameters of shared components, which initialize only once at the beginning of the first task. As mentioned earlier, these new task-specific parameters and shared parameters are optimized according to the training data of the new task for another batch of training epochs. We used the RMSProp optimizer with the hyper-parameter values (such as learning rate and the number of

training epochs per task) described in Table 2. In our experiments, we conservatively have LASEM take a single M-step per E-step by setting $numMSteps = 1$.

## B CATASTROPHIC FORGETTING OF LASEM

We investigated catastrophic forgetting of LASEM in addition to mean peak per-task accuracy. The catastrophic forgetting ratio is shown in Figure 7. The catastrophic forgetting ratio, proposed in Lee et al. (2019), measures the ability of the lifelong learning algorithm to maintain its performance on previous tasks during subsequent learning. A low ratio indicates that there is negative reverse transfer from new tasks to previously learned tasks, and so the learner experiences catastrophic forgetting. A ratio greater than 1 can be interpreted as positive backward transfer. As depicted in Figure 7, LASEM is able to retain the performance of previous tasks compared to transferring at all CNN layers and transferring at specific CNN-layers for all tasks (using a static transfer configuration).

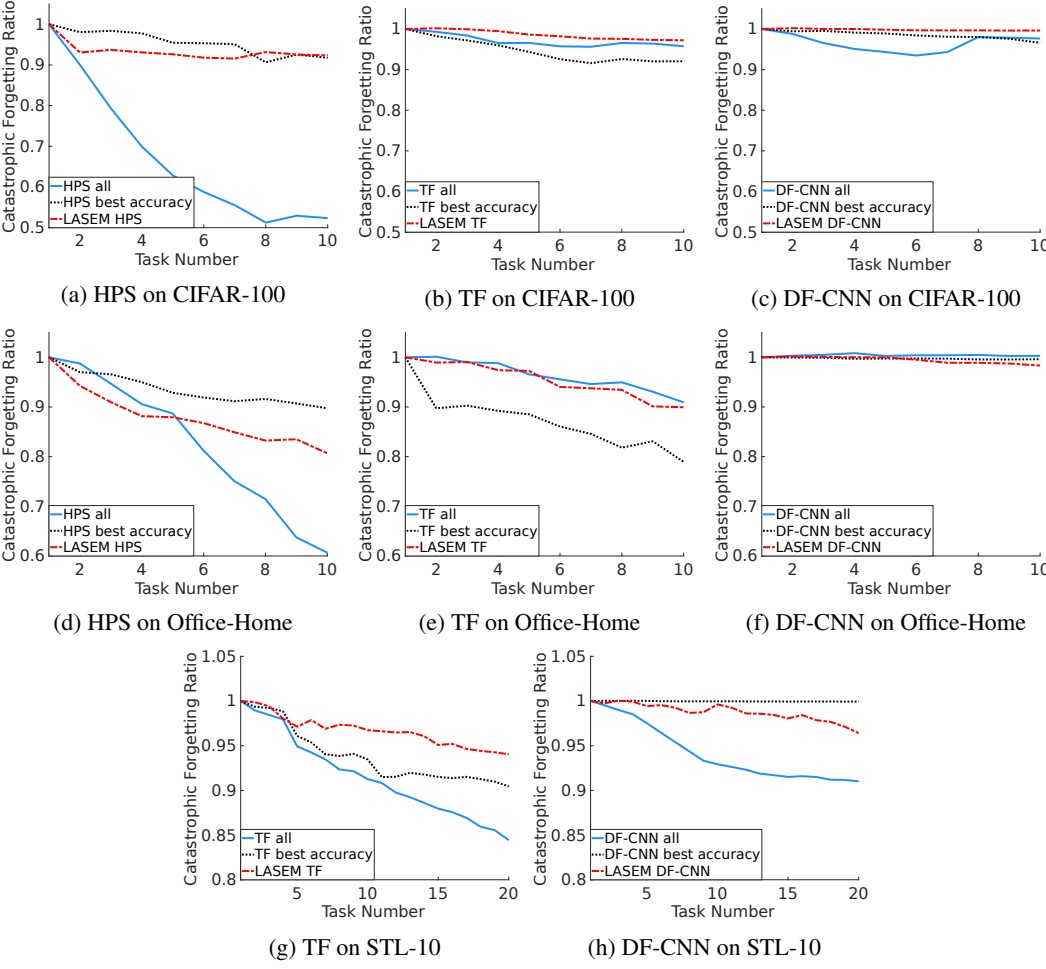

Figure 7: Catastrophic forgetting ratio of transfer at all CNN layers (blue), best static transfer configuration (black) and LASEM (red), exhibiting the benefit of LASEM. Note that the y-axis range differs for each data set.

## C COMPARISON TO BRUTE-FORCE CONFIGURATION SEARCH

We compared LASEM to the performance of task-wise brute-force search over transfer configurations, shown in Table 3. Here, "task-wise brute-force search over transfer configurations" refers to the method of training every transfer configuration and choosing the best configuration for each task,

| Architecture | LASEM | Transfer All Layers | | Brute-force Search | |
|---|---|---|---|---|---|
| | Accuracy (%) | Accuracy (%) | Rel. Time | Accuracy (%) | Rel. Time |
| CIFAR-100 (10 Tasks) | | | | | |
| HPS | $39.3 \pm 0.1$ | $24.7 \pm 0.6$ | 0.78 | $40.4 \pm 0.3$ | 6.55 |
| TF | $38.4 \pm 0.5$ | $36.3 \pm 1.0$ | 0.64 | $39.9 \pm 1.1$ | 8.81 |
| DF-CNN | $42.0 \pm 0.6$ | $36.3 \pm 1.3$ | 0.59 | $42.6 \pm 0.7$ | 9.45 |
| Office-Home (10 Tasks) | | | | | |
| HPS | $58.4 \pm 0.9$ | $54.9 \pm 0.7$ | 0.72 | $59.4 \pm 0.2$ | 4.72 |
| TF | $59.1 \pm 1.0$ | $56.2 \pm 0.7$ | 0.66 | $58.7 \pm 0.3$ | 5.22 |
| DF-CNN | $59.5 \pm 1.1$ | $49.1 \pm 0.6$ | 0.61 | $58.8 \pm 0.3$ | 4.04 |

Table 3: Comparison of test accuracy and training time for the same epochs to brute-force configuration search, $\pm$ 95% confidence interval. Training time shown above is ratio to training time of LASEM. *Relative time* greater than 1.0 means that training the model is slower than our approach.

thereby allowing task models to use different transfer configurations. The accuracy of brute-force search is almost indistinguishable from LASEM at 95% confidence, but it requires at least $3\times$ additional time for training. This result shows that LASEM achieves approximately the best accuracy of dynamic transfer configuration with a boost in training speed.

## D  LASEM-DISCOVERED TRANSFER CONFIGURATIONS

Figure 8 shows the most frequent transfer configurations as well as the proportion of the time each layer was chosen to be transfer-based or task-specific. For CIFAR-100 and Office-Home, there is tendency of transferring top layers more than bottom layers. However, interesting observation is that non-tree structures, such as Alternating $\{2, 4\}$ and sharing middle layers [0,1,1,0], are often chosen. This contradicts the assumption of a tree structure made often by related research, and supports the consideration of more complex transfer configurations for diverse tasks.

The top eight most-chosen configurations of STL-10, unlike the other datasets which employed deep nets with fewer CNN layers, plateau with a peak less than 10%. This is likely due to the smaller number of STL-10 tasks, more flexible (deeper) network, and a much larger number of possible transfer configurations than in the other two experiments. The tensor factorization model for STL-10 seems to prefer transfer at higher layers to transfer more than lower layers, while the preference for DF-CNN is more varied.

## E  LASEM FOR TRANSFER OVER GROUPS OF LAYERS

As discussed in Section 3, it is a well-known problem in neural architecture search that the search over layer-based transfer configurations requires time exponential to depth of the network $d$. One common technique to compensate for this problem, as used by other methods (Pham et al., 2018; Liu et al., 2018), is to search over groups of layers instead of individual layers, thereby reducing the size of the search space. LASEM easily supports this same technique by redefining the transfer configuration space $\mathcal{C} = \{0, 1\}^d$ to be binary indicators over a partition $\mathcal{P}$ of the set of layer indices $\{1, \ldots, d\}$, where the cardinality $|\mathcal{P}| \ll d$. Consequently, this reduces the search space from $2^d$ to $2^{|\mathcal{P}|}$. Most naturally, the partition $\mathcal{P}$ should ensure that either adjacent layers (e.g., $\{\{1, 2\}, \{3, 4\}, \{5, 6\}\}$) or nearby[1] layers (e.g. $\{\{1, 3\}, \{2, 4\}, \{5\}, \{6\}\}$) are grouped together.

We evaluated this variation of LASEM in lifelong learning scenarios using the STL-10 data set. Different from the aforementioned experiments using STL-10, this experiment consisted of 15 five-way classification tasks by random selection of the classes. We sampled only 10% of the given training data and split it into training and validation sets with the ratio 1:1. For this scenario, we trained a DF-CNN transferring at all layers and layer-based LASEM on a DF-CNN with 9 convolutional layers. The group-based LASEM used the partition $\{\{1, 2, 3\}, \{4, 5, 6\}, \{7, 8, 9\}\}$, splitting 9 convolutional layers into three groups of three adjacent layers.

---

[1]This pattern of grouping nearby layers together, instead of only adjacent, may allow more flexible adaptation of transferred knowledge to individual tasks, similar to the Alternating configuration explored in Section 2.

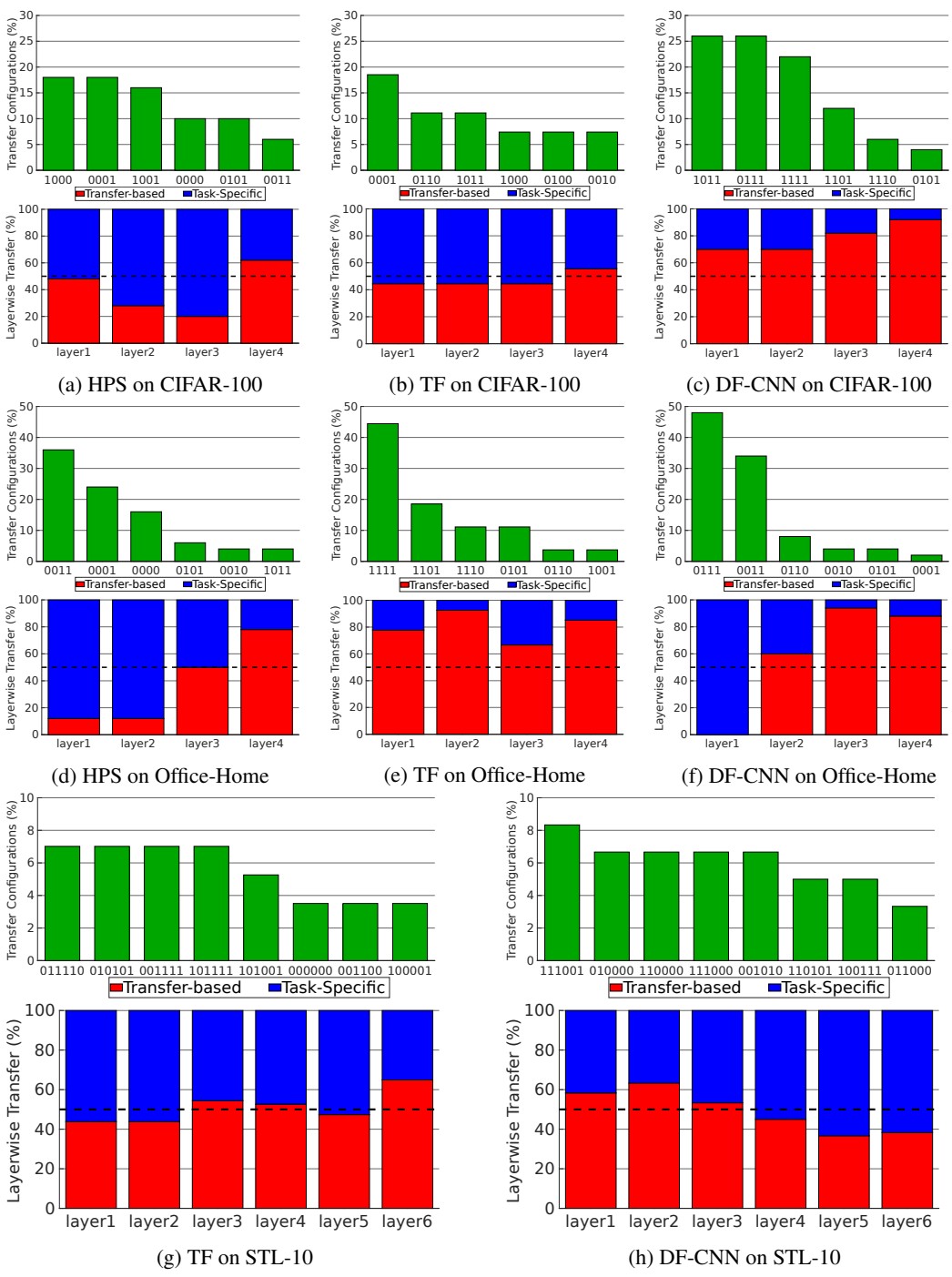

Figure 8: (Top) Histogram of the most-selected configurations (i.e., the binary vectors $c_t$, where 1 denotes that a CNN layer employs transfer). (Bottom) The fraction of the time each layer was selected to be transfer-based (red) or task-specific (blue).

The DF-CNN achieved a mean task-wise best accuracy of $45.4 \pm 0.4$ with a training time of $7.58 \times 10^4$ seconds. The layer-based LASEM DF-CNN exceeded the capacity of our computing source (an Intel core i7 workstation with dual 1080 Ti GPUs). However, the group-based LASEM DF-CNN achieved a mean accuracy of $46.0 \pm 0.7$ % in $7.02 \times 10^4$ seconds. This showcases LASEM's ability to support group-based transfer configurations in addition to layer-based configurations.

## F  MEMORY USAGE COMPARISON

In this section, we analyze LASEM's memory requirements to show that it is approximately equivalent to the other methods considered in the paper. Let the base learner require $O(A)$ non-transfer-based task-specific storage or $O(B)$ transfer-based task-specific storage with $O(S)$ shared knowledge. LASEM shares network parameters across transfer configurations to minimize memory, so the *current* task model stores two parameter sets at a cost of $O(A + B)$. Earlier task models require only $O(max(A, B))$ storage, yielding a total memory requirement of $O(S + (T + 1)max(A, B))$ for T tasks when the base learner constructs one network per task. Compared to this, the model with the best static transfer configuration requires $O(max(A, B))$ additional storage per task, resulting in a total memory requirement of $O(S + T\ max(A, B))$. Brute-force search over transfer configurations requires at least $O(2\ max(A, B))$, one for parameters of the best configuration and the other for parameters of the current training, yielding $O(S + (T + 1)max(A, B))$. Hence, LASEM's memory requirements are approximately equivalent to the alternative methods. The memory requirement may differ from the above analysis according to the base lifelong learning architecture used (such as HPS or a modular network), but if that base learner requires $O(T)$ memory for $T$ tasks, LASEM needs only $O(T + 1)$ memory.

