# OpenReview forum: "Sharing Less is More: Lifelong Learning in Deep Networks with Selective Layer Transfer"
_ICLR.cc/2021/Conference — Reject_

### Official Review · AnonReviewer2 · 2020-10-27
**Nicely executed paper**

**Rating:** 7
**Confidence:** 4

**Review:**

Summary: authors look into life long learning setting (when task arrive one after another) and try to understand which layers of the source model need to be reused (transfered) and which should be re-learnt. Authors argue that this decision should be task specific, and come up with an algorithm (EM like) that for each task derives indices of the layers to reuse and also updates reusable and non reusable layers.

Overall, this paper is really well executed. Easy to follow, enough of background and motivation is provided, and the algorithm for the most part is clear and intuitive.
My main complaint is that it is somewhat thin on the experiments.

Additional questions/comments:
1) How is update 13 from Algo 1 happening? (how P(C|data) is updated? is it  working by recalculating the counts from step (1) (so it is not a smooth function that is differentiable and updated via optimization)
2) In Algo 1: while IsMoreTrainingDataAvailable is not clear to me. I assume that you do several epochs over task specific data, don't you? Also do you monitor anyhow EM convergence?
3) Experiments: One baseline would be useful to just randomly select c_t for each task and see how it does. EM is pretty expensive
4) Finally, I feel that related work should really go after the into

---

> ### Author Response · Authors · 2020-11-18
> **Thanks for your feedback. We have added clarifications about our algorithm**
>
> We would like to thank the reviewer for the feedback. We are encouraged that the most of our paper is clear and intuitive to the reviewer. We answer questions that the reviewer specified below.
>
> **Q1. Details of update on line 13 of Algorithm 1:** Equation 1 is the detail of the function PriorUpdater() in line 13 of Algorithm 1. It counts the number of mini-batches for the current task (line 2-14 of Algorithm 1). Only the M-step, line 11, requires gradient computation, so the prior is not a smooth function.
>
> **Q2. Details of IsMoreTrainingDataAvailable and convergence:** Yes, we do several epochs over data of a specific task. Algorithm 1 considers the general setting in which data of any task can be given to the learner at each timestep. The idea is that at each timestep, the learner gets whatever data is available regardless of tasks, so the function getNextTrainingData() returns a task index t as well as training data $X_{new}$ and $y_{new}$ for the learner to know what the current task is. As long as more data is available, the function isMoreTrainingDataAvailable() will be true, and the learner can then incorporate available data. In the experiment of the paper, we imposed additional constraints on the setting so that tasks remain the same along the specific amount of time or epochs.
>
> We considered lifelong learning scenarios in which a learner has no control over tasks, so the current task can be switched to another one without convergence of the learning algorithm. Because of this setting, we don’t require EM convergence, but it is possible to monitor the convergence by checking the probability weight over transfer configurations.
>
> **Q3 Additional Ablative Baseline:** That’s an interesting suggestion, and what essentially occurs at the start of LASEM for a new task (when the transfer configuration is essentially random). But, it’s likely to yield performance somewhere around the means of the black boxes of Figure 3, which is clearly exceeded by just a bit of search over the configurations.  Additionally, during the early stage of LASEM, both the transfer configuration and the layer parameters are not converged, so converging the transfer configuration immediately via random sampling as you suggest in the ablative baseline wouldn’t likely accelerate convergence.
>
> **Q4 Related Work Placement:** Thanks for your suggestion on the paper structure. As you can tell, the paper structure is a bit atypical (but earlier feedback said that it worked well for this paper), so it’s good to have another viewpoint.

---

### Official Review · AnonReviewer4 · 2020-10-28
**Review for Sharing Less is More: Lifelong Learning in Deep Networks with Selective Layer Transfer**

**Rating:** 6
**Confidence:** 3

**Review:**

This paper studies the problem of lifelong learning of a sequence of tasks by selectively transferring knowledge of some layers across tasks. The proposed approach, LASEM, uses EM to dynamically adjust transfer configuration between tasks by performing architecture search. The authors present results in benchmark datasets for continual learning.

As strengths of the paper I would remark:
-	The paper is well-motivated and demonstrates experimentally why simply transferring all layers does not lead to the best transfer configuration.
-	The proposed approach in section 3 is clearly explained and the main design choices are well argued. In general, the paper is well-written and easy to follow.
-	The paper includes thorough results of the computational complexity of the proposed method, which could be a concern given the proposed approach of maintaining both shared and specific sets of parameters for each task.

The weaknesses I observe in this paper are:
-	I am concerned on the extensibility of the proposed method to a large number of tasks, which I would think would pose a challenge in the proposed setting similar to how it does in multitask learning. What would be the impact of a large number of tasks, specifically on the set of shared parameters? The experiments provide results for a limited number of tasks (up to 20).

Questions for authors:
- Please address the question regarding the effect of number of tasks on the proposed approach.

---

> ### Author Response · Authors · 2020-11-18
> **Thanks for your feedback. We have added explanation on your concern about the extensibility of our LASEM method**
>
> We would like to thank the reviewer for the feedback. We are encouraged that the reviewer found our paper well-motivated. The reviewer raised a question about the extensibility of the proposed method to a large number of tasks.
>
> We’re following experimental protocols (including datasets and tasks) used by previous work, and you’re certainly correct that the number of tasks is limited.  This is largely to permit numerous evaluations for statistical purposes within a reasonable time.  However, our approach should scale to numerous tasks.
>
> Our algorithm aggregates only the information of the current task and resets when the task is changed. Therefore, the computational requirement of LASEM remains the same regardless of the number of tasks. The memory requirement of task models according to the number of tasks is directly related to the lifelong learning architecture that LASEM is applied to.
>
> As the number of tasks grows larger, so too often does the task distribution broaden, and so the capacity of the shared tensors would likely need to be set correspondingly larger to handle the increased breadth of the task models.  This observation is less relevant to our proposed LASEM algorithm, and more relevant to the base lifelong learners that it operates upon.
>
> Note that the extensibility of LASEM is highly correlated to the depth of the network as other neural architecture search methods do. We adopted a solution of previous works of neural architecture search for networks with many layers, and the result is described in Appendix E.

---

### Official Review · AnonReviewer3 · 2020-10-28
**Review of the Paper**

**Rating:** 4
**Confidence:** 5

**Review:**

-Summary-
The paper proposes a method for selective weight sharing per layer during continual learning. The authors show observations that sharing all layers can not be optimal for lifelong learning. Hence, they adopt a layerwise transfer configuration vector which decides activated layer-sharing at specific tasks. The problem is solved by EM algorithm-based approach.

-Pros-
- Observations are reasonable and give many inspirations for solving continual learning issues and developing the existing methods.
- The paper is well written and easy to follow.
- The point of view connected to neural architecture search is understandable.
- The model outperforms old baselines.

-Cons-
- The problem is task-incremental which clearly gives task oracle during training and inference. Recent continual learning works obviously tackle class-incremental learning problems that are more challenging and applicable to a broader area [1].
- Baselines are too weak. If the paper targets task-incremental learning problems, the authors should compare their methods with recent works, rather than with 3-5 past years' works. I recommend to include further strong baselines like [2,3,4].
- The strengths of the methods may not be impressive on modern deeper networks, like ResNet-50. Since it requires massive computation time. The paper didn't show the results/analysis of modern deep architectures.
- The model inevitably requires much capacity for not-shared task-specific layers. But the authors didn't include it.

-Comments-
- Why do the authors only use a fraction of datasets?


===============
- [1] van de Ven, Gido M., and Andreas S. Tolias. "Three scenarios for continual learning." arXiv preprint arXiv:1904.07734 (2019).
- [2] Titsias, Michalis K., et al. "Functional Regularisation for Continual Learning with Gaussian Processes.", ICLR 2020.
- [3] Yoon, Jaehong, et al. "Scalable and Order-robust Continual Learning with Additive Parameter Decomposition.", ICLR 2020.
- [4] Davide Abati, et al.  "Conditional Channel Gated Networks for Task-Aware Continual Learning.", CVPR 2020.

---

> ### Author Response · Authors · 2020-11-18
> **Thanks for your feedback. We have added clarifications about our algorithm and your concerns**
>
> We would like to thank the reviewer for the feedback. We are encouraged that the most of our paper is clear and intuitive to the reviewer. We answer the reviewer’s questions.
>
> **Con1. Class-incremental learning problem is more challenging than task-incremental problem:** We agree that lifelong learning problems without task indices are more challenging, but task-incremental learning is far from solved. There are many applications which require task-incremental learning, and so work in that topic is certainly relevant and still current. In the future, we do aim to extend this work to handle learning scenarios even when task information is not available to the learner.
>
> **Con2. Baselines:**  We are implementing the suggested baselines and running experiments. We will incorporate the comparison with the baselines during revision.
>
> **Con3.  Application to deeper networks:** This work proposes the method to find useful layers to be shared between tasks, so we’re dependent on the network architectures used by existing lifelong learning methods. Deeper networks like ResNet-50 show great results on visual domain problems, but these architectures are not designed for knowledge transfer between tasks. If one ResNet-50 with multiple output heads, for instance, is used in a lifelong learning scenario, it is conceptually the same as hard-parameter sharing architecture because of the explicit re-use of parameters.
>
> It is an astute observation that the reviewer pointed out the difficulty of applying neural architecture search (NAS) to a deeper network. It is a common issue in the NAS literature, and previous works remedy this issue by applying NAS methods over groups of layers instead of individual layers. We used this solution to apply LASEM to the network with 9 convolutional layers to share, and the details and experimental results are described in Appendix E. Lifelong learning experiments (repeated many times to get proper statistics) take a while, and so we simply don’t have the computational resources to try it over larger networks such as ResNet-50.
>
> **Con4. Unspecified capacity for non-shared task-specific layers:** We’ve tried to understand your concern, but it is unclear.  Would you mind restating it in more detail?
>
> Appendix A specifies the sizes of both shared and non-shared layers, and we theoretically analyzed the memory usage of the method with respect to the number of tasks (including the task-specific layers) in Appendix F.  The concern about the non-shared layer storage seems more an aspect of the base lifelong learner than our LASEM algorithm that operates on that base learner. Since this paper focuses on a multi-model formulation of lifelong learning, the total number of additional parameters (for the shared layers and the transfer configuration storage) is a constant as the number of tasks grows.  The storage for non-shared layers does grow proportionally to the number of tasks, but such per-task parameters are typical for such multi-model lifelong learners.  Please see the references for the base lifelong learning algorithms used in the paper.
>
> **Question. Reason for using a fraction of datasets:** We followed previous work on this (see experiment details in Section 2 and Appendix A), using a fraction of the datasets to evaluate lifelong learning performance in a low-data regime. In the low-data regime, it is important to transfer useful knowledge of previously learned tasks due to the lack of training data. Therefore, using a fraction of datasets makes comparison between knowledge transfer methods clear, and allows direct comparison to previous work.

---

> > ### Comment · AnonReviewer3 · 2020-11-23
> > **Feedback from authors response**
> >
> > **Con1. & Con2.**
> > - I agree that task-incremental learning is not solved. That's the reason that recent continual learning work proposes a compatible approach like FRCL [1]. Further, **I have waited** for comparison with **recent strong task-incremental learning methods** that I suggested, **rather than 3-5 past years work**. But the author doesn't give any experimental evaluations. To this end, **I strongly doubt whether the proposed method is competitive or not.**
> >
> > [1] Titsias, Michalis K., et al. "Functional Regularisation for Continual Learning with Gaussian Processes.", ICLR 2020.
> >
> > **Con3.**
> >
> > - NAS is one of the rapidly growing topics of the community. Recent NAS literature already shows the impressive performance using further deeper search space. If the proposed method is hard to perform with further deeper networks because of its inherent methodological limitation, **that can be a strong weakness** of the method, since most of the cutting-edge methods on various domains **do not use simple CNN** with a few of conv layers. **If the DART-based approach is not fit on deeper networks, the authors should consider using better recent methods.**
> >
> > **Con4.**
> >
> > - **The method is not scalable** that the total number of parameters is proportional to the number of tasks. Since the continual learning model supposes to learn a number of various tasks during sequential training, the **linear increase of the model capacity** is a critical issue [2,3,4].
> >
> > [2] Rusu, Andrei A., et al. "Progressive neural networks." arXiv preprint arXiv:1606.04671 (2016).
> >
> > [3] Li, Xilai, et al. "Learn to grow: A continual structure learning framework for overcoming catastrophic forgetting." ICML 2019.
> >
> > [4] Lee, Soochan, et al. "A Neural Dirichlet Process Mixture Model for Task-Free Continual Learning." ICLR 2020.
> >
> > **Additional**
> > - The methods are **performed well with a small and single dataset**, but there is no evidence that the proposed method consistently outperforms baselines (recent baselines like the ones that I referred in the initial review) under **heterogeneous datasets** like [5, 6]. This is crucial to validate the transferability across different datasets.
> >
> > [5] Serra, Joan, et al. "Overcoming catastrophic forgetting with hard attention to the task." ICML 2018.
> >
> > [6] Zhai, Xiaohua, et al. "A large-scale study of representation learning with the visual task adaptation benchmark." arXiv preprint arXiv:1910.04867 (2019)
> >
> > ----------------
> > I thoroughly read the authors' responses and constructive feedback from other reviewers. As other reviewers mentioned, the paper has great potential with an impressive contribution to the CL community. But the author didn't compare the **essential baselines during the rebuttal period**. The method has inherent **scalability limitations** in terms of the applicability of modern networks and linear increase of the model capacity. These are critical to continual learning. And overall baselines and references of the paper are **not up to date** even though both continual learning and NAS literature are rapidly developing topics. To this end, I strongly recommend to develop the paper to strengthen its intriguing contributions.

---

> > > ### Author Response · Authors · 2020-11-24
> > > **Comparison to the new suggested baselines**
> > >
> > > We would like to thank the reviewer for the additional feedback. We are sorry for taking so long into the rebuttal period to report the comparison to the suggested baselines, but the experiments took a while to run -- we wanted to ensure a fair comparison to existing work through extensive hyperparameter search on these new baseline methods.
> > >
> > > **Con2. Results on New Suggested Baselines:** We empirically evaluated a conditional channel-gated network (GateNet) [1] and an additive parameter decomposition (APD) network [2] on our Office-Home experiment (extension of Section 4.2).
> > >
> > > We tried numerous sets of hyper-parameters (i.e. architecture of gating modules, sparsity loss weight) for the GateNet, but the best performance of the GateNet is far from the level comparable to other baselines and LASEM (despite accuracy better than random guessing $\simeq$ 7.69%). Since we weren’t able to get comparable performance with the hyper-parameters sets we tried for this method, we are going to continue the hyper-parameter search in the hopes that we’ll be able to make this baseline perform better.
> > >
> > > APD achieved comparable task-wise accuracy to ProgNN and LASEM DF-CNN. However, APD showed its weakness when it comes to the knowledge retention of the earlier tasks. According to the catastrophic forgetting ratio (greater than or close to 1.0 meaning better retention), APD loses the performance of the previously learned tasks more than ProgNN and LASEM DF-CNN.
> > >
> > > | Method                    |Accuracy (%)                        | Forgetting Ratio                    |
> > > |---------------------------|------------------------------------|--------------------------------------|
> > > |GateNet                    |$$22.07 \pm 0.87$$            | $$0.890 \pm 0.052$$           |
> > > |APD                            |$$\mathbf{59.58 \pm 0.45}$$            | $$0.828 \pm 0.028$$           |
> > > |LASEM DF-CNN        |$$\mathbf{59.45 \pm 1.10}$$            | $$\mathbf{0.983 \pm 0.013}$$            |
> > >
> > > **Con3 & Con4. Scalability of LASEM:** For clarification, the proposed LASEM method is for the search of transfer configuration, not the knowledge transfer architecture between tasks. For instance, the methods which the reviewer referred to in reviews combine two lifelong learning problems: (1) how to transfer knowledge between tasks and (2) which granularity of knowledge (i.e. layer) is useful for transfer between tasks. LASEM, however, proposes a solution for the second problem, so it requires a base lifelong learning architecture (such as HPS and DF-CNN) for the empirical evaluation.
> > >
> > > The lack of results applying LASEM to a deeper network is because of the computational resources that we have. As explained in the Section 3 (page 5) and Appendix E, LASEM is applicable to a deeper network as other NAS methods are, but our resources cannot train even ResNet-50 in a lifelong learning scenario within the reasonable amount of time.
> > >
> > > The required model capacity being linear to the number of tasks originates from the lifelong learning architecture, not LASEM. In experiments, we applied LASEM to models that have one network per task because it is straightforward to visualize the relation between task models and transfer configurations. LASEM is applicable to architectures like modular networks which have no linear growth of model capacity, since LASEM itself is an algorithm to find transfer configurations according to data-driven task relationships.
> > >
> > > **Additional:** We appreciate the reviewer for pointing out additional (extended, heterogeneous datasets) experimental settings that we could run for an extended version of this paper. We will be sure to explore those in future work.
> > >
> > > [1] Davide Abati, et al. “Conditional Channel Gated Networks for Task-Aware Continual Learning.”, CVPR 2020.
> > >
> > > [2] Yoon, Jaehong, et al. “Scalable and Order-robust Continual Learning with Additive Parameter Decomposition.”, ICLR 2020.

---

### Official Review · AnonReviewer1 · 2020-11-01
**Algorithm needs more clarification**

**Rating:** 6
**Confidence:** 2

**Review:**

This paper introduces a lifelong learning algorithm across multiple tasks that automatically learns which layers need to be optimized using an EM learning strategy for each task. In the expectation step, the algorithm updates the next best configuration and in the M step, it optimizes the model parameters.

Some details of the algorithm are not explained well. It could help a lot if the authors could provide the objective function they are trying to optimize. As examples: (1) The motivation behind Equation 3 is not clear at all. Based on Equation 3, It seems that the choice of different configurations is mini-batch dependent and not task dependent. I have a difficult time to understand why this is the case. I might have misunderstood the algorithm but providing the objective function helps a lot. (2) It is not clear why n_{c_i} in Equation 1 is the number of previous mini-batches for which C_{(t)} is the most probable configuration. It makes sense to use a soft version of this (sum of probabilities) to have it more compatible with the rest of the algorithm. Again, understanding what objective function the authors are targeting to optimize helps to understand this better too.

Even though, some aspect of the proposed algorithm is not clear, the algorithm works in practice and outperform multiple suggested baselines.

Minor type: Page 5: 5% percent  5%

---

> ### Author Response · Authors · 2020-11-18
> **Thanks for your feedback. We have added clarifications about our algorithm**
>
> We would like to thank the reviewer for the feedback. We are encouraged that most of our paper is clear and intuitive to the reviewers. We answer the questions regarding the algorithm of our proposed method.
>
> **Q1. Clarification of equation 3:** The choice of configurations is task-dependent. Although the selection may be updated by each mini-batch, the selection is based on the aggregate updates over all mini-batches. The selection of transfer configurations is based on their relevant weight, which is computed by the posterior (equation 2) combining information from the current mini-batch (likelihood) and task-wise history (prior, equation 1).
>
> The objective function of the optimization is log-likelihood $P(y_{new}|X_{new}, c_{i})$. Equation 3 is simply the gradient updates of the objective for all configurations weighted by the estimated posterior of those configurations. So, you can think of it as the expected gradient update based on the distribution over the configurations. Consequently, Equation 3 is the M-step of Algorithm 1 (line 11). Please note that the two equations in Equation 3 are almost identical except that one is for the parameters of the shared layers and the other is for the parameters of task-specific layers.
>
> **Q2. Soft version of counting configurations for equation 1:** When developing this approach we tried two formulations for the prior probability of the transfer configuration: (1) the method described in the paper using the number of mini-batches (Equation 1), and (2) the sum of probabilities of the soft version of Equation 1, as you suggested.  These two methods were not different statistically in an empirical evaluation on CIFAR100 and Office-Home, so we chose the simpler of the two methods to include in the paper.  We’ll make a note of this in the revision.
>
> Thanks for finding the typo!

---

### Author Response · Authors · 2020-11-24
**Paper Revision**

To All ICLR reviewers:
Thank you again for your detailed feedback in the reviews.  We have updated our submission based on your suggestions, with all changes colored in blue.  The major changes include:
1. Addressing alternative priors to Eqn 1 and clarifying the objective. (For Reviewer 1; pg 4)
2. Clarifying the lack of EM convergence requirements. (For Reviewer 2; pg 5)
3. Clarifying Algorithm 1. (For Reviewer 1 and Reviewer 2; pg 5)
4. Adding a comparison to a new suggested baseline of APD-Net (Yoon et al., 2020), showing that our approach has equivalent accuracy with less catastrophic forgetting. (For Reviewer 3; pg 7)
5. Added additional suggested related work (Multiple reviewers; pg 8)
6. Clarifying that memory scalability of our LASEM approach is only O(T+1) for a base learner that requires O(T) memory for T tasks. (For Reviewer 3 and Reviewer 4; pg 17)
7. Clarifying lifelong learning metrics, especially accuracy $\rightarrow$ peak per-task accuracy.

Please note: As requested by Reviewer 3, we attempted a comparison against GateNet (Abati, et al. CVPR 2020), but despite a moderate hyperparameter search over 1 week on a verified implementation, were unable to achieve results comparable to the other methods on Office-Home, yielding an accuracy of only 22.07% +/- 0.87 with a forgetting ratio of 0.89 +/- 0.052. We will continue the hyperparameter search for GateNet, and if we are able to get it to reach comparable accuracy to the other methods, will add it to Table 1.

---

### Decision · Program_Chairs · 2021-01-07
**Final Decision**

**Decision:**

Reject

**Comment:**

The reviewers enjoyed reading about an interesting take on lifelong learning, encapsulating an EM methodology for selecting a transfer configuration and then optimizing the parameters. R3 made valid concerns regarding comparison with previous, recent work. R2 also would prefer to see more thorough experiments (ideally in settings where multiple tasks exist, as also commented by R4). During the rebuttal phase the authors made a good effort to run additional experiments which cover the related work aspect better. These experiments and the overall paper were discussed extensively among reviewers after the rebuttal phase.

In the discussions, the reviewers agreed that an interesting idea can be publishable even if it does not achieve SOTA results in all scenarios, as long as it brings new perspectives and shows at least comparable results. However, in the particular case of this paper, there exist remaining concerns regarding the usefulness and applicability of the method. Specifically, the paper could benefit from a more convincing demonstration about how the method can scale (e.g. R3 and R4’s comments), especially since training time and model capacity are important factors to consider for practical continual learning scenarios. Furthermore, it is not clear how the proposed method can be used in combination with other machine learning tools within a continual learning application, for example by leveraging modern deep architectures or by complementing existing adaptive knowledge approaches (as discussed by R3).

Although the opinions of the reviewers are not fully aligned, this borderline paper seemed to lack an enthusiastic endorsement by a reviewer to compensate for the concerns discussed above and the relatively weak experimental results. Therefore I recommend rejection.